# Genotype Sequencing and Phylogenetic Analysis Revealed the Origins of *Citrus Yellow Vein Clearing Virus* California Isolates

**DOI:** 10.3390/v16020188

**Published:** 2024-01-26

**Authors:** Yong-Duo Sun, Raymond Yokomi

**Affiliations:** United States Department of Agriculture, Agricultural Research Service, San Joaquin Valley Agricultural Sciences Center, Parlier, CA 93648, USA

**Keywords:** *Citrus yellow vein clearing virus* California isolate, genome sequencing, spatiotemporal phylogenetic analysis, Bayesian evolutionary inference, virus evolution

## Abstract

The *Citrus yellow vein clearing virus* (CYVCV) causes a viral disease that has been reported in some citrus-growing regions in countries in Eurasia including Pakistan, India, Türkiye, Iran, China, and South Korea. Recently, CYVCV was detected in a localized urban area in a town in the middle of California’s citrus-growing region and marks the first occurrence of the virus in North America. CYVCV has been reported to be spread by aphid and whitefly vectors and is graft and mechanically transmitted. Hence, it is an invasive pathogen that presents a significant threat to the California citrus industry, especially lemons, which are highly symptomatic to CYVCV. To elucidate the origin of the CYVCV California strain, we used long-read sequencing technology and obtained the complete genomes of three California CYVCV isolates, CA1, CA2, and CA3. The sequences of these isolates exhibited intergenomic similarities ranging from 95.4% to 97.4% to 54 publicly available CYVCV genome sequences, which indicated a relatively low level of heterogeneity. However, CYVCV CA isolates formed a distinct clade from the other isolates when aligned against other CYVCV genomes and coat protein gene sequences as shown by the neighbor network analysis. Based on the rooted Maximum Likelihood phylogenetic trees, CYVCV CA isolates shared the most recent common ancestor with isolates from India/South Asia. Bayesian evolutionary inferences resulted in a spatiotemporal reconstruction, suggesting that the CYVCV CA lineage diverged from the Indian lineage possibly around 1995. This analysis placed the origin of all CYVCV to around 1990, with South Asia and/or Middle East as the most plausible geographic source, which matches to the first discovery of CYVCV in Pakistan in 1988. Moreover, the spatiotemporal phylogenetic analysis indicated an additional virus diffusion pathway: one from South Asia to China and South Korea. Collectively, our phylogenetic inferences offer insights into the probable dynamics of global CYVCV dissemination, emphasizing the need for citrus industries and regulatory agencies to closely monitor citrus commodities crossing state and international borders.

## 1. Introduction

The *Citrus yellow vein clearing virus* (CYVCV) presents a pressing quarantine concern regarding the transport of citrus commodities and international trade, as well as the unwitting and illegal movement of infected citrus budwood or propagations. The manifestations of CYVCV disease exhibit significant variations contingent upon citrus varieties and prevailing environmental conditions [1]. Lemon (*Citrus limon*) and sour orange (*C. aurantium*) trees are highly symptomatic, while a broad range of other citrus cultivars, though susceptible, remain generally asymptomatic [2,3]. CYVCV-symptomatic citrus trees display stunted growth, diminished citrus yields, yellow vein clearing, the water-soaked appearance of veins on the adaxial side, leaf deformities, intermittent ringspots, and venial necrosis [1,2,3]. No effective management strategies have been found to counteract the deleterious impact of CYVCV, since the citrus host range is wide and insect vectors are common in citrus orchards. In addition to infecting most citrus species, cultivars, and hybrids, CYVCV has been reported to be transmitted to some non-citrus hosts, including wild grapevine (*Vitis vinifera*), common bean (*Phaseolus vulgaris*), cowpea (*Vigna unguiculata*), common mallow (*Malva sylvestris*), quinoa (*Chenopodium quinoa*), and some other herbaceous species [4,5,6]. 

CYVCV is a member of the *Alphaflexiviridae* virus family, the *Mandarivirus* genus, and constitutes a positive-sense flexuous RNA virus [7,8]. It is noteworthy that CYVCV shares a high genome similarity, of approximately 74% sequence identity, with *Indian citrus ringspot virus* (ICRSV)*,* another member of the *Mandarivirus* genus [9]. To date, only a handful of CYVCV isolates have been subjected to complete sequencing [3,10,11]. The viral genome encompasses approximately 7530 base pairs and encodes six predicted open reading frames (ORFs). ORF1 encodes a solitary polyprotein with four constituent subunits, namely methyltransferase, oxygenase, RNA helicase, and RNA-dependent RNA polymerase. ORF2 to ORF6 encode triple gene block gene 1 (TGB1), TGB2, TGB3, coat protein (CP), and a nucleic acid-binding protein [1,8]. Our understanding of the functional properties of CYVCV-encoded proteins remains limited. CP has been identified as an RNA silencing suppressor [12] and has been linked to the severity of symptoms in citrus. Subsequent research has elucidated the CP’s interaction with the 40S ribosomal subunit protein S9-2, whose transient accumulation in the host impedes the CP’s silencing suppressor activity [13]. 

CYVCV can be transmitted through grafting and mechanical means and is naturally vectored by at least three aphids (*Aphis spiraecola*, *A. craccivora*, and *A. gossypii*) and the citrus whitefly, *Dialeurodes citri* [6,14]. The initial report of yellow vein clearing disease was in lemon and sour orange in Pakistan in 1988 [15]. Since then, it has been reported in various locales, including Türkiye, India, Iran, China, and South Korea [2,16,17,18]. The rapid proliferation of CYVCV in China since 2009 has resulted in substantial losses in lemon production where the disease incidence was high [19]. In 2022, during a routine multi-pest survey conducted by the California Department of Food and Agriculture (CDFA), CYVCV-infected citrus trees were identified in localized urban properties in the city of Tulare, California, United States of America (USA), although surveys of nearby citrus orchards indicate no spread yet to commercial citrus [10]. The United States Department of Agriculture, Animal, and Plant Health Inspection Service also tested samples from infected trees and verified the diagnosis of CYVCV. The CDFA currently designates CYVCV as a pest of high concern (Pest rating A).

The evolutionary dynamics of CYVCV remain enigmatic. This knowledge gap is, in part, attributable to the limited availability of CYVCV sequence data. Additionally, the virus’s evolutionary and ecological dynamics may have developed on different time scales, a phenomenon observed in other plant viruses [20]. In a 2019 report, a phylogenetic tree constructed using the maximum likelihood method delineated CYVCV isolates into two clusters. CYVCV isolates from India, Türkiye, and Pakistan coalesced in one group, while those from China formed another [9]. Recent developments, including CYVCV’s expansion into citrus-growing regions in California and South Korea, have drawn attention to the virus’s origin, global transmission, and spread [10]. As more CYVCV genotypes are identified, comparative genomics would contribute to a deeper understanding of CYVCV’s etiology, relationships, and evolution. In this study, employing state-of-the-art long-read sequencing technology, we successfully obtained the complete genome sequences of three novel CYVCV California isolates. Through genotypic clustering and Bayesian evolutionary analysis, we elucidate the spatiotemporal and phylodynamics of CYVCV on a global scale. Though the CYVCV isolate dataset of our analysis remains relatively shallow, this inference would provide critical information for citrus stakeholders to monitor the dispersal of the virus crossing state and international borders.

## 2. Materials and Methods

### 2.1. Sample Collection, RNA Extraction, and Real-Time Quantitative Polymerase Chain Reaction (RT-qPCR)

Citrus budwood from known CYVCV-positive citrus trees from the city of Tulare, California were collected and graft-propagated to a variety of citrus cultivars in a containment greenhouse. CYVCV source cultivars included Eureka lemon, mandarin, red grapefruit, kumquat, and makrut lime (*Citrus hystrix*). The grafted plants were maintained in an air-conditioned greenhouse at the San Joaquin Valley Agricultural Sciences Center in Parlier, California. The original host of CYVCV CA1 and CA2 was Eureka lemon, while CA3 was from makrut lime. Total RNA from CYVCV the isolates were as extracted by Trizol Reagent (ThermoFisher Scientific, Waltham, MA, USA) from leaves exhibiting symptoms of CYVCV. 

A duplex RT-qPCR for the simultaneous detection of CYVCV and the citrus *Nad5* gene as an internal quality control was employed. Specifically, the RT-qPCR reaction took place in a 10 µL reaction volume composed of 2 µL of RNA template, 5 µL of 2× reaction buffer, 0.4 µL each of CYVCV forward primer (5′-AAA TCC ATT AAC ACA GTG ACC TTC C-3′) and reverse primer (5′-AAC TCC TGA CAG TGC TCC AA-3′), 0.1 µM of a CYVCV-specific 6-FAM/BHQ-1 labeled TaqMan probe (5′d FAM-CGTCGTTGCCAAGACACGCCA-BHQ-1), 0.4 µL each of Nad5 forward primer (5′-GATGCTTCTTGGGGCTTCTTKTT-3′) and reverse primer (5′-ACATAAATCGAGGGCTATGCGGATC-3′), and 0.1 µM of a Nad5-specific VIC/QSY labeled TaqMan probe (5′d VIC-CAT AAG TAG CTT GGT CCA TCT TTA TTCCAT-QSY), along with 0.2 µL of iScript advanced reverse transcriptase and 0.9 µL of double-distilled water. This mixture was placed into a PCR plate, with cycling conditions encompassing reverse transcription at 50 °C for 5 min, initial denaturation at 94 °C for 2 min, followed by 40 cycles of denaturation at 94 °C for 10 s, and annealing/extension at 60 °C for 40 s [21]. RNA samples at a concentration of 10 ng/µL were tested in triplicate.

### 2.2. CYVCV Genome Sequencing

A conserved region at the 5′ end, identified through alignment with other reported CYVCV isolates, served as the basis for designing a virus-specific 5′ race primer (5′-GGTTAGTGGTATTGCCCTGTT-3′). As for the 3′ race-specific primer, an oligo(dT) primer was employed. The amplicons generated from the 5′ and 3′ race PCRs were subjected to purification and subsequent cloning into the pGEM-T easy vector (Promega Corp., Madison, WI, USA). At least three constructed vectors were sequenced (Plasmidsaurus, Eugene, OR, USA) to obtain the sequences of the CYVCV 5′ and 3′ termini.

Using these 5′ and 3′ termini sequences, the complete genome sequences were amplified for CYVCV CA isolates using the Q5 high-fidelity enzyme (New England Biolabs Inc., Ipswich, MA, USA) and virus-specific PCR primers (5′ primer-GAAAAGCAAACATAACCAACACACACCC; 3′ primer-CAGAAAATGGAAACTGAAAGCCTGAATATTT). This yielded a 7.5 Kb PCR amplicon which was sequenced with the latest long-read sequencing technology from Oxford Nanopore Technologies (ONT, Plasmidsaurus, Eugene, OR, USA). The fully assembled genome sequences were annotated and deposited in GenBank under the accession numbers OR037276.1 (CYVCV CA1), OR670060 (CYVCV CA2), and OR6700601 (CYVCV CA3).

### 2.3. Nucleotide Diversity Analysis

To analyze the nucleotide diversity, the complete genomes of 57 CYVCV isolates were downloaded from the NCBI Virus Database and aligned using the NCBI Multiple Sequence Alignment Viewer. The Virus Intergenomic Distance Calculator (VIRIDIC) was employed to generate a heatmap utilizing default settings that incorporated intergenomic similarity values and alignment indicators [22].

### 2.4. Recombination Analysis

The recombination analysis of 57 different CYVCV isolates and 79 coat protein gene sequences (either full/978 bps or partial gene sequences of Iran isolates ranging from 498 bps to 614 bps) was conducted using the Recombination Detection Program v4.56 (RDP4) software [23]. The software utilized various algorithms, including RDP, GENECONV, CHIMAERA, MAXCHI, BOOTSCAN, SISCAN, and 3SEQ, each of which identified putative recombination events, major and minor parents, and breakpoints. Recombination events detected by at least four different methods were considered.

### 2.5. Construction of Non-Rooted Phylogenetic Neighbor Network Tree and Rooted Maximum Likelihood Tree

The complete genomes of 53 nonrecombinant CYVCV isolates and 79 coat protein gene sequences (either full/978 bps or partial gene sequences of Iran isolates ranging from 498 bps to 614 bps) were acquired from GenBank, NCBI, and aligned using MEGA 11 with the MUSCLE algorithm. The construction of a neighbor network was executed and subsequently modified utilizing SplitsTree 4, with 1000 bootstrap replicates [24]. A maximum likelihood (ML) tree was estimated using two different models, a non-clock (unconstrained) generalized time-reversible (GTR) and gamma substitution model and Hasegawa–Kishino–Yano (HKY) and gamma substitution model using IQ-Tree [25]. The trees generated with two different models share the same topology. The maximum likelihood midpoint root tree was visualized and adjusted using FigTree v1.4.4. 

### 2.6. Bayesian Evolutionary Inference

To assess the degree of divergence signal accumulated over the sampling time interval, the CYVCV CP sequence data were used, and an exploratory linear regression approach was conducted. Initially, a maximum likelihood (ML) tree was estimated under a non-clock (unconstrained) HKY and gamma substitution model using IQ-Tree [26]. Root-to-tip divergences were plotted as a function of sampling time, employing a root maximized to yield the Pearson product moment correlation coefficient through TempEst (formerly known as Path-O-Gen) [27].

Subsequently, a time-calibrated phylogenetic tree was reconstructed using a Bayesian statistical framework from the software package BEAST v1.10.4 [28]. Different recombinations of nucleotide substitution models, such as those of HKY and Gamma and GTR and Gamma; clock models, such as strict clock and relaxed clock; and tree priors, including Expansion growth, Exponential growth, Constant size, and Logistic growth, were tested. Upon comparison, the substitution model employed was HKY and Gamma, the clock type was set as strict clock, and a Logistic growth tree providing the best fit. The length of the Markov chain Monte Carlo (MCMC) chain was set as 20,000,000. BEAST employed MCMC integration to average over tree space, weighting each tree proportionally to its posterior probability. MCMC chains were visually checked by Tracer v1.6 and posterior parameters from tree samples were summarized via Treeannotator. A consensus tree was visualized and modified with FigTree v.1.4.3.

### 2.7. Biogeographic Analyses

Ancestral geographic ranges at each node were reconstructed using Statistical–Dispersal Vicariance Analysis (S-DIVA) and Bayesian Binary MCMC (BBM) analysis through the program Reconstruct Ancestral States in Phylogenies (RASP) [29]. Four distribution ranges were defined, based on geographic proximity, as South Asia (India and Pakistan), East Asia (China and South Korea), Middle East (Türkiye), and North America (USA).

## 3. Results

### 3.1. Field Survey and CYVCV qPCR Detection

A diverse collection of CYVCV was acquired from known infected trees (*C. limon*, *C. hystrix*, *C. reticulata*, *C. paradisi*, *Fortunella* sp.) from different properties in Tulare, CA, USA and propagated in the greenhouse (Figure 1A). Graft propagations of sour orange (*C. aurantium* L.) and Eureka Lemon (*C. limon*) exhibited clear CYVCV symptoms of vein clearing, water soaking, and leaf distortion (Figure 1B), whereas virus propagations in mandarin (*C. reticulata*), *C. macrophylla*, Duncan grapefruit (*C. paradisi*), Madam Vinous sweet orange (*C. sinensis* (L.) Osbeck), and S1 citron (*C. medica* L.) remained asymptomatic (not shown). Systemic CYVCV infection was confirmed by RT-qPCR using United States Department of Agriculture, Animal, and Plant Health Inspection Service Plant Protection and Quarantine (USDA-APHIS-PPQ)-approved CYVCV primers. An intriguing observation emerged when analyzing Eureka lemon plants: virus titers were notably higher in flowers compared to stems and leaves, despite the absence of obvious growth defects or disease symptoms in flowers (Figure 1C).

### 3.2. The Whole Genome Sequences of CYVCV CA Isolates and Nucleotide Diversity Analysis

Three CYVCV isolates were selected that represented different CYVCV sources propagated from six different infected field trees collected over a 5.2 sq km area of Tulare. The complete genomes of these three isolates were sequenced and designated as CYVCV CA1 (Accession number OR037276.1), CYVCV CA2 (Accession number OR670060), and CYVCV CA3 (Accession number OR670061). This addition of three new CYVCV California isolates brings the total number of reported CYVCV whole genome sequences to 57 (Table 1). These genomes of the three California isolates consisted of 7530 nucleotides (nt), excluding the 3′ poly A tail, and harbored six open reading frames (ORFs). It is noteworthy that the genome size of CYVCV isolates ranged from 7528 nt to 7531 nt due to insertion/deletion mutations at positions 20, 29, 30, and 6127 nt (Appendix A). Multiple sequence alignments displayed base-pair differences with variations, which indicated that CYVCV CA isolates exhibited a relatively high divergence compared to the reference sequence CYVCV CQ isolate (Accession number NC_026592.1) from China (Figure 2A). Sequence identity analysis unveiled that the reported global CYVCV isolates share a high sequence similarity, ranging from 95.1% to 100%, indicating a relatively low level of heterogeneity (Figure 2B). Within this spectrum, the sequences of these CYVCV CA isolates exhibited intergenomic similarities ranging from 95.4% to 97.4% to 54 publicly available CYVCV genome sequences. In contrast, the CYVCV CA in-group isolates exhibited a relatively high sequence identity, reaching 99.6%. Notably, two pairs of CYVCV isolates share a hundred percent identity, as the CYVCV AY112 (Accession number MW429487.1) is identical to CYVCV AY204 (Accession number MG878869.1), and CYVCV CQ isolate NC_026592.1 is the same as another CYVCV CQ isolate KP313240.1.

### 3.3. CYVCV Genotype Groups

Phylogenetic analysis using a neighbor-net reconstruction of CYVCV complete genomes unveiled two major genotype groups (Figure 3A). All CYVCV isolates from China and South Korea formed a major group termed the “East Asia group”, which further comprised eight subgroups, including the South Korea subgroup (SK1) and China subgroups 1–7 (C1–C7), named in chronological order of sample collection (Table 1). Other CYVCV isolates were grouped into one major genotype, the “South Asia, Middle East, and North America group”, encompassing isolates from India, Pakistan, Türkiye, and California/USA (Table 1). However, unlike the East Asia group, limited data availability hindered sub-clustering within this second major group. Nucleotide diversity analysis suggested closer genetic relationships between CYVCV isolates from India, Pakistan, Türkiye, and California.

### 3.4. Dissecting the Origin of CYVCV CA Isolates Based on Whole Genome Data

To ascertain the origin of CYVCV CA isolates, we initially assessed whether these isolates were the result of recombination. Among the 57 submitted CYVCV genome sequence data, four recombination events were found as previously detected in a small-scale study of *Mandarivirus* [9]. The CYVCV GX-STJ isolate (Accession number KX156742.1), CYVCV CQ-PO isolate (Accession number KX156735.1), CYVCV AY221 isolate (Accession number MW429491.1), and CYVCV PALI isolate (Accession number KT696512.1) were identified as recombinants. The analysis revealed no recombination events among CYVCV CA isolates (Appendix A). Subsequently, a maximum likelihood phylogenetic analysis was performed using 53 CYVCV genome sequences, excluding the 4 identified as recombinants among the 57 submitted. This analysis also grouped CYVCV isolates into two major groups (Figure 3B). CYVCV isolates from China and South Korea clustered together, while isolates from other regions formed a separate group. Notably, CYVCV CA isolates shared the most recent common ancestor with an India CYVCV RMGI isolate (Accession number KT696511.1). This analysis suggested that CYVCV likely originated from India, with the India CYVCV ECAI isolate (Accession number KT696510.1) connecting directly to the root of the tree. Isolates from India, Pakistan, Türkiye, and the United States appeared to be more closely related to the ancestor of CYVCV compared to isolates from East Asia. It is worth mentioning that no whole genome sequences were obtained from Iran, despite reports of CYVCV presence in 2007.

### 3.5. CYVCV Grouping and Phylogeny upon CP Sequences

Though no whole genome sequences of CYVCV Iran have been reported, 12 CYVCV Iran isolate partial CP sequences, ranging from 498 bps to 614 bps, are available in the public database. To test the possible origin of CYVCV CA isolates from Iran, the 12 CYVCV Iran isolate CP partial sequences, along with 67 full length CP sequences, making a total of 79 CP sequences, were retrieved from GenBank, NCBI (Table 2) and were employed to generate a phylogeny tree. Upon alignment, sequences of MN547330.1 and MN547329.1 (Iran), AWJ64286.1 and WBG00067.1 (China), WBG00091.1 and WBG00085.1 (China), ASK39538.1 and ASK39436.1 (China), ASK39490.1 and ASK39454.1 (China), ASK39526.1 and ASK39520.1 (China), AJO26403.1 and YP009124992.1 (China), ASK39496.1 and ASK39454.1 (China) are identical, but were kept for subsequent analysis. Neighbor-net analysis, based on CP sequences, classified CYVCV isolates into four major groups: East Asia (China and South Korea), Middle East (Iran and Türkiye), South Asia (India and Pakistan), and North America (United States). These groups were named according to their geographical distribution (Figure 4A). Notably, CYVCV CA isolates formed a distinct clade, but relatively close to the CYVCV South Asia and Middle East groups. Similar to the phylogeny generated using the whole CYVCV genome data, CYVCV CA isolates were separate from the CYVCV East Asia group. Since no evidence of recombination was found in the CP datasets (not shown), a maximum likelihood rooted tree was further constructed and revealed that CYVCV CA isolates/North America groups shared the most recent common ancestor with several India isolates (Accession numbers: KT696516.1, KT696518.1, KT696520.1, AOO32386.1) (Figure 4B). Thus, consistent with results obtained from whole genome sequences, phylogenetic analysis based on CP sequences suggested that CYVCV CA isolates may have originated from India.

### 3.6. Inferring the Spatiotemporal Origin of CYVCV

Next, we employed another phylogenetic inference method, the Bayesian phylodynamic framework, to estimate the CYVCV origin. Compared to the maximum livelihood phylogenetic analysis, Bayesian phylodynamic inference could provide information regarding the virus spatiotemporal dispersal. The 53 non-recombinant CYVCV whole genome sequences, collected at various time points over 31 years, were employed. Root-to-tip analysis using TempEst indicated a moderate temporal structure in the dataset, with the maximum correlation coefficient being equal to 0.621 (Appendix A). To maximize the correlation coefficient, the ancestor traces tool was applied to identify and eliminate the problematic sequences. Only sequences that drew a green line from the virus to the point on the regression line, indicating where the immediate ancestor should lie, were kept. With optimization, 38 CYVCV sequences were chosen for further analysis and the correlation coefficient between the divergence from the root and time of sampling was improved to 0.8002, displaying a strong positive relationship (Appendix A).

A maximum clade credibility (BBM) tree from the Bayesian molecular clock analysis of the 38 selected CYVCV whole genome dataset estimated the origin of CYVCV from its progenitor to approximately 1990 (Figure 5). This estimated origin period is close to the first report of CYVCV in 1988 and the first disclosure of CYVCV genome data (CYVCV IS) in 1992. The origin of all five CYVCV strains from the South Asian and Middle East region can be traced back before 2000, which implies that the CYVCV population may have circulated in this hypothetical restricted region for many years before its global spread to East Asia and the USA. The most recent common ancestor of the CYVCV CA lineage appeared around 2018, four years prior to its documented discovery in California. Additionally, the CYVCV CA lineage shared a common ancestor with an Indian lineage, suggesting an origin around 1995 (Figure 5). Thus, in line with geographical distribution and historical records, it is plausible that CYVCV CA isolates originated in South Asia, potentially India.

Next, a phylogenetic tree showing the ancestral distribution ranges based on the BBM model was reconstructed by the program Reconstruct Ancestral States in Phylogenies (RASP) (Figure 6A) and, subsequently, a virus global diffusion map was suggested (Figure 6B). In line with the previous data, the most likely distribution region of the most recent common ancestor of CYVCV CA isolates is in India/South Asia. Moreover, the spatiotemporal phylogenetic analysis suggested an additional virus diffusion pathway: from South Asia to East Asia. This suggests that the CYVCV China lineage shares the most common ancestor with a virus isolate identified from Pakistan/South Asia. CYVCV GJ isolates from South Korea were inferred to have originated from CYVCV isolates in Yunnan, China.

## 4. Discussion

CYVCV, as an emerging viral disease in citrus, has become a major concern for some citrus-producing regions worldwide. Reported lemon yield losses of up to 50–80% in China underscore its economic impact [30]. Over the past few decades, CYVCV isolates have been reported in seven countries: India, Pakistan, China, South Korea, Iran, Türkiye, and, most recently, California, USA [2,16,17,18]. Although the first observations of CYVCV-infected trees date back to 1988 in Pakistan, the first complete genome sequencing of CYVCV was not reported until four years later [15]. In the case of Iran, CYVCV was reported in 2017, but, to date, no whole genome sequences of CYVCV isolates from Iran have been made publicly available [18]. The incursion of CYVCV into California, USA was documented in 2022; however, the genome sequence and molecular characteristics of CYVCV CA isolates have not been documented until this report. Our study leveraged cutting-edge long-read sequencing technology to obtain and annotate three CYVCV CA isolates from Tulare County, CA, to expand the CYVCV dataset and deepen our understanding of virus divergence, which may become useful for disease management purposes. 

The complete genome sequences of CYVCV CA isolates exhibit the typical genome organization of the *Mandarivirus* genus, ranging from 7529 to 7531 nt, excluding the 3′-poly (A) tail [1]. The three CYVCV isolates obtained in this study encompassed 7530 nt each. Global CYVCV isolates, when subjected to genome-wide comparison, displayed a high sequence similarity, ranging from 95.1% to 100%, indicating a limited degree of heterogeneity. Typically, the RdRP region of RNA viruses harbors functional domains of replicase proteins and is prone to nucleotide variability. The absence of proofreading activity in RNA polymerases of RNA viruses presents the potential for rapid evolution, genetic variability, and adaptation to new environmental conditions due to high mutation rates, resulting in the generation of variable populations [9,31]. The mechanism behind this observation requires further exploration, as it may shed light on the replication mechanism of CYVCV.

Studying the genetic and molecular diversity of viral pathogens contributes to a deeper understanding of virus ecology, evolution, and biology. In this context, our study aimed to investigate recombination and population dynamics using various statistical algorithms. The data obtained in this study revealed low levels of genetic diversity, but still supported new knowledge of the evolutionary relationships within the virus populations. Neighbor network analysis, considering full genome sequences of CYVCV isolates from different countries, indicated that CYVCV isolates from India, Pakistan, and Türkiye were more closely related than those from China and South Korea (Figure 3B). Beside the whole genome sequence, the CP region of RNA viruses is vital for species demarcation, assessing genetic diversity, and developing immunodiagnostics [32,33]. A previous CP analysis revealed that viruses in the *Mandarivirus* genus shared a common structural core and evolutionary origin [34]. By analyzing the CYVCV CP dataset, including a few partial CP sequences from Iran, since whole genome sequence has not yet been reported, we found that CYVCV isolates could be categorized into groups based on their geographic distribution, including South Asia, East Asia, Middle East, and North America. Maximum likelihood analysis of CP sequences indicated a similar outcome when compared to the whole genome analysis: CYVCV CA isolates diverged from isolates in India, a part of the South Asia region that has reported a substantial number of different CYVCV isolate sequences (Figure 4A,B). As the CYVCV genome pool expands, the boundaries of CYVCV genotypes will likely become more clearly defined.

In addition to elucidating the spatiotemporal scale of plant virus evolution, molecular sequence analyses can explore spatial population structures and provide insights into the dissemination dynamics responsible for the current geographic distribution of plant viral lineages. Therefore, it is not surprising that plant virus epidemiology has started to incorporate statistical inference methods that combine temporal and spatial dynamics in a phylogenetic context [28,33]. For example, Bayesian phylogeographic methods have been applied to reconstruct the spatiotemporal history of *Tomato yellow leaf curl virus* spread and diversification. This analysis suggested that the virus likely originated in the Middle East during the first half of the 20th century [35]. Another example is *Cassava mosaic-like virus*, responsible for severe crop losses in sub-Saharan Africa, which was estimated to have originated in mainland Africa in the late 1930s, with subsequent introductions to the southwest Indian ocean islands between 1988 and 2009 [36]. Similarly, there is *Maize streak virus* (MSV), which has caused severe epidemics in maize-growing regions of Africa. Bayesian spatiotemporal reconstructions indicated southern Africa as the most probable origin of MSV at the beginning of the 20th century [37]. 

In this study, Bayesian evolutionary analysis based on CYVCV whole genome sequences suggested that CYVCV CA isolates may have diverged from South Asia, specifically India, around 1995. However, it is essential to recognize that spatiotemporal phylogenetic analysis has its limitations. First, the CYVCV isolate dataset remains relatively shallow, particularly in regions such as South India, Middle East, and North America. Additionally, the detection of plant viruses in perennial hosts is often delayed, as it takes time for symptoms to manifest. Moreover, symptom variations in different citrus varieties can further complicate disease diagnosis. For instance, while CYVCV is generally associated with vein clearing symptoms in sensitive cultivars, it can also produce ringspot symptoms on Kinnow mandarin and sweet orange, similar to those caused by ICRSV. Delayed detection in some commercial citrus varieties in India with ringspot symptoms was due to a lack of information about CYVCV’s ability to cause such symptoms [2,17,19]. Therefore, caution must be exercised when interpreting phylogenetic relationships with limited biological data or a lack of regional perspectives during a plant virus outbreak. Incorporating additional characteristics to support phylogenetic interpretation will likely yield more reliable inferences [38]. For instance, it has been suggested that the host ecology determines the dispersal patterns of rice yellow mottle virus [20]. 

## Figures and Tables

**Figure 1 viruses-16-00188-f001:**
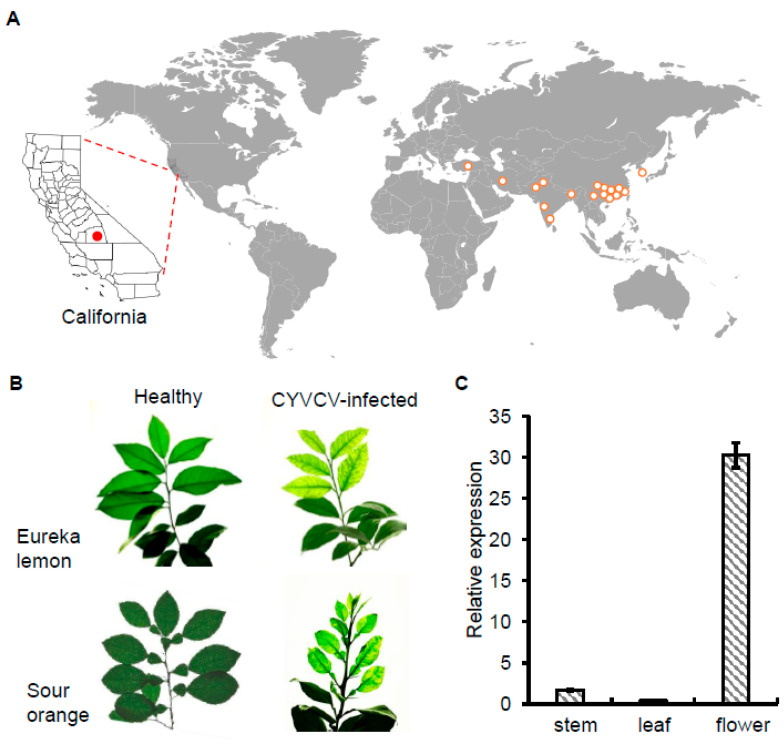
*Citrus yellow vein clearing virus* (CYVCV) outbreaks and traits. (**A**). Global distribution of CYVCV. The global occurrence of CYVCV, except South Korea, was derived from data reported in the European and Mediterranean Plant Protection Organization Global Database, accessed in August 2023.The emerging of CYVCV in Tulare County, California, United States was marked wit a red dot. All the others were marked with orange circles. (**B**). A typical branch of Eureka lemon (*Citrus limon*) and Sour orange (*C. aurantium*) trees showing a typical CYVCV-induced yellow vein clearing phenotype in the greenhouse. (**C**). RT-qPCR data shows the distribution patterns of CYVCV in lemon stem, leaf, and flower. X axis depicts different citrus organs, while Y axis stands for the relative expression of *Citrus yellow vein clearing virus*. The citrus Nad5 gene was used as an internal control. Three biological tests were repeated, and a similar trend was obtained.

**Figure 2 viruses-16-00188-f002:**
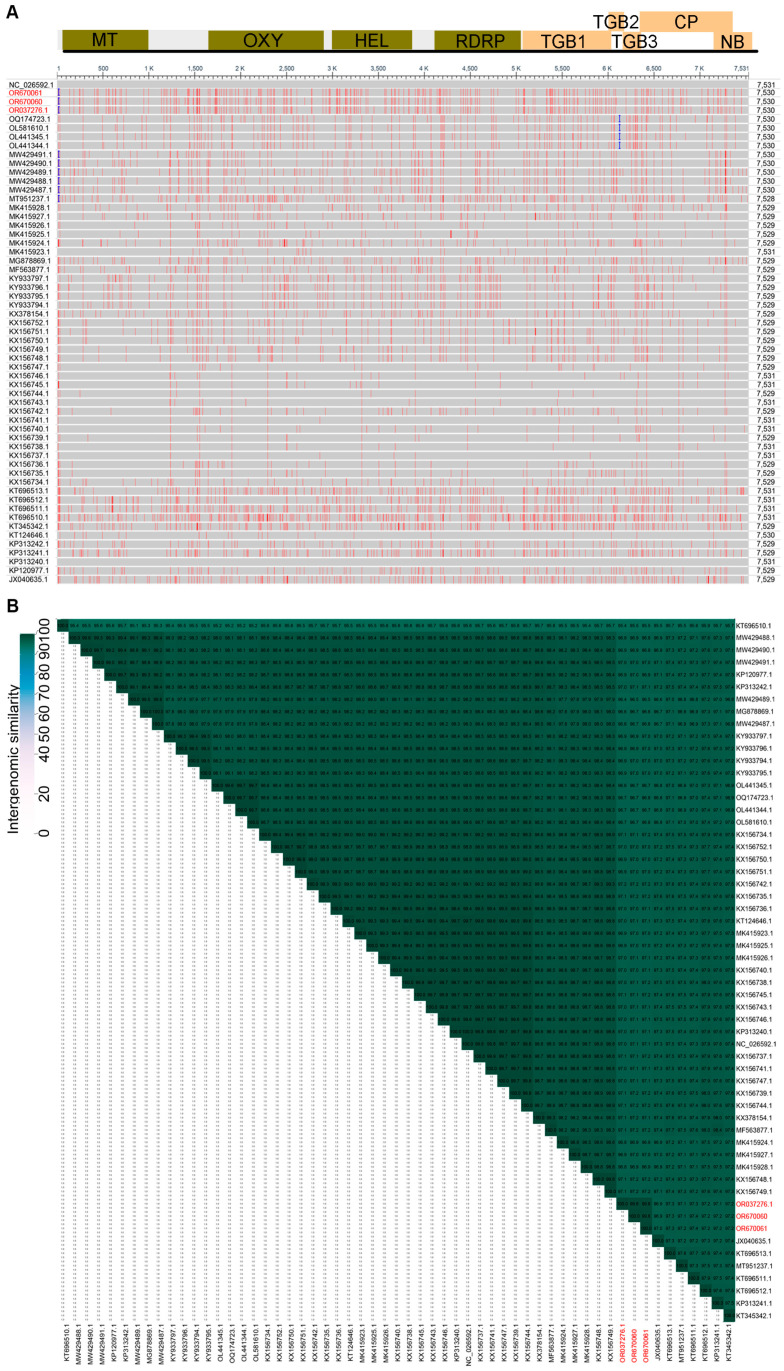
*Citrus yellow vein clearing virus (CYVCV)* nucleotide diversity. (**A**). Multiple alignment reveals the base-pair with frequency-based difference, which was marked with red in the figure. (**B**). VIRDIC generated heatmap incorporating intergenomic the right half, the color-coding allows a rapid visualization of the intergenomic similarity of CYVCV genomes. The CYVCV CA isolates were marked with red.

**Figure 3 viruses-16-00188-f003:**
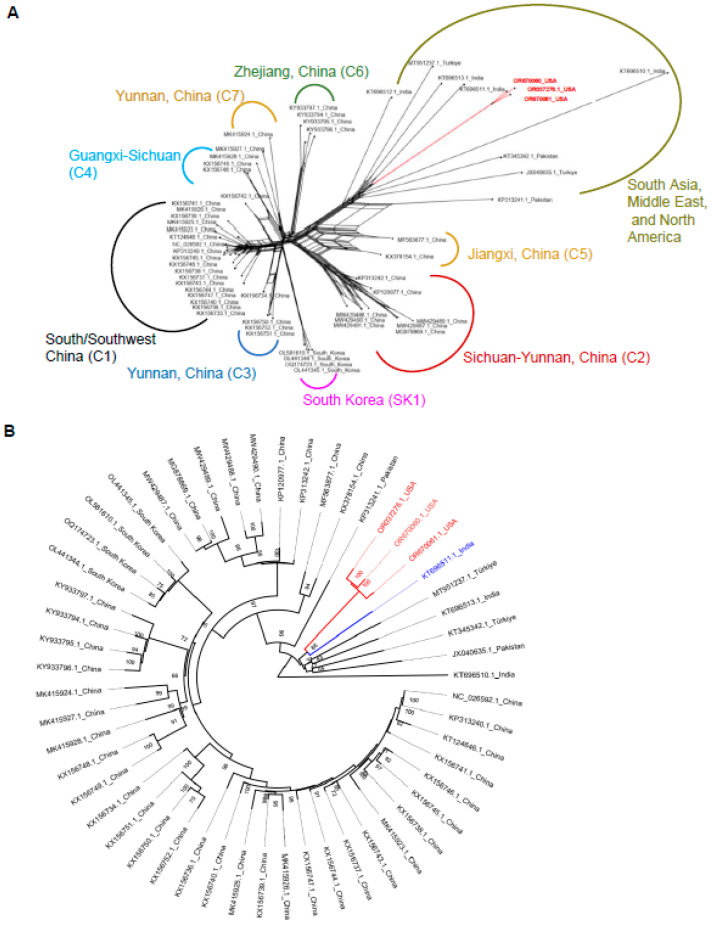
Construction of non-rooted and rooted phylogenetic tree upon *Citrus yellow vein clearing virus (CYVCV)*. (**A**). Neighbor network reconstruction of the complete genomes of 57 CYVCV isolates. The clade of CYVCV CA isolates is marked red. (**B**). Maximum isolates identified as recombinant. The clade of CYVCV CA isolates is marked with red. An India virus isolate in the same clade with CYVCV CA isolates is labeled with blue. Rooting method: Midpoint. Node labels display: posterior probabilities.

**Figure 4 viruses-16-00188-f004:**
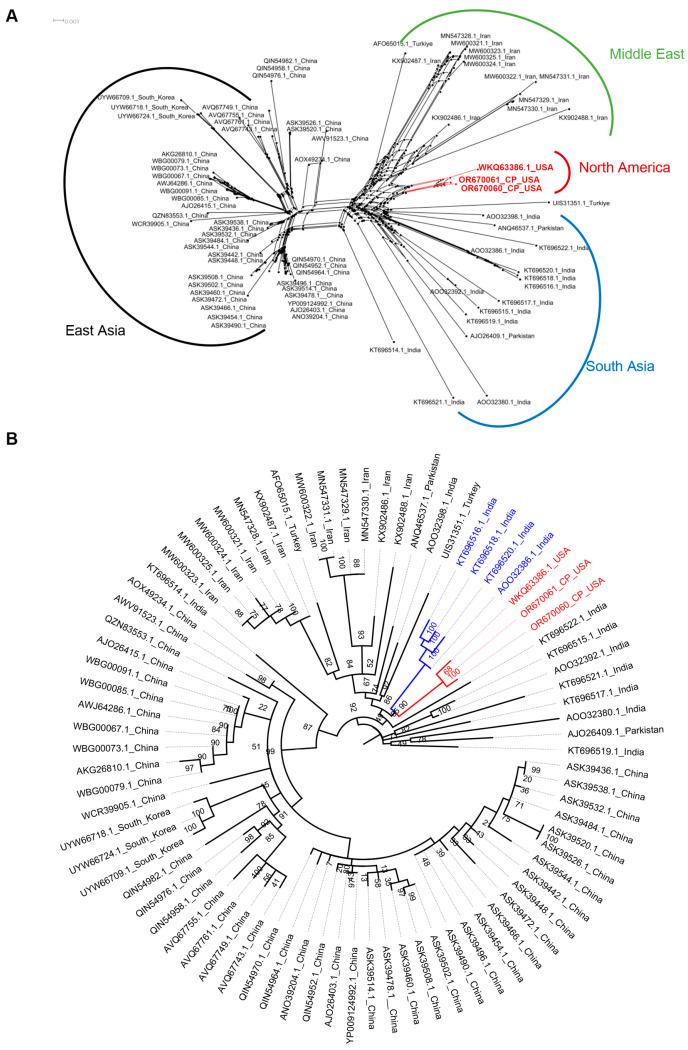
Construction of non-rooted phylogenetic tree upon *Citrus yellow vein clearing virus (CYVCV)* coat protein (CP) sequence. (**A**). Neighbor network reconstruction of the CP sequence of 79 CYVCV isolates. The clade of CYVCV CA isolates is marked red. (**B**). Construction of a Maximum likelihood tree to trace the most recent ancestor of CYVCV CA isolates upon the 79 CP sequence examined in this study. The clade of CYVCV CA isolates is marked red. Four India virus isolates in the same clade with CYVCV CA isolates are labeled with blue. Rooting method: Midpoint. Node labels display: posterior probabilities.

**Figure 5 viruses-16-00188-f005:**
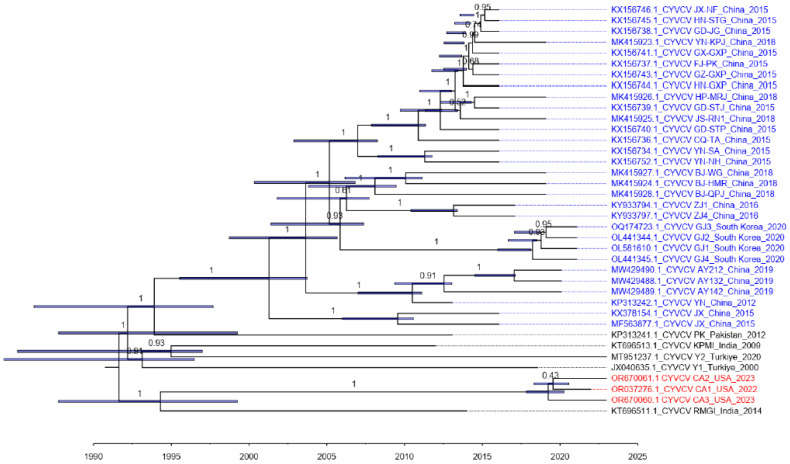
Maximum clade credibility BEAST tree reconstructed upon the 38 selected *Citrus yellow vein clearing virus* whole genome sequences. The different color of the taxa indicates different geological distribution: Blue: East Asia (China and South Korea); Red: North America (United States); Black: South Asia (India and Pakistan) and Middle East (Türkiye). Branch labels display posterior. Node bars display the 95% highest posterior density of the node heights. The tree was visualized and modified with FigTree v.1.4.3.

**Figure 6 viruses-16-00188-f006:**
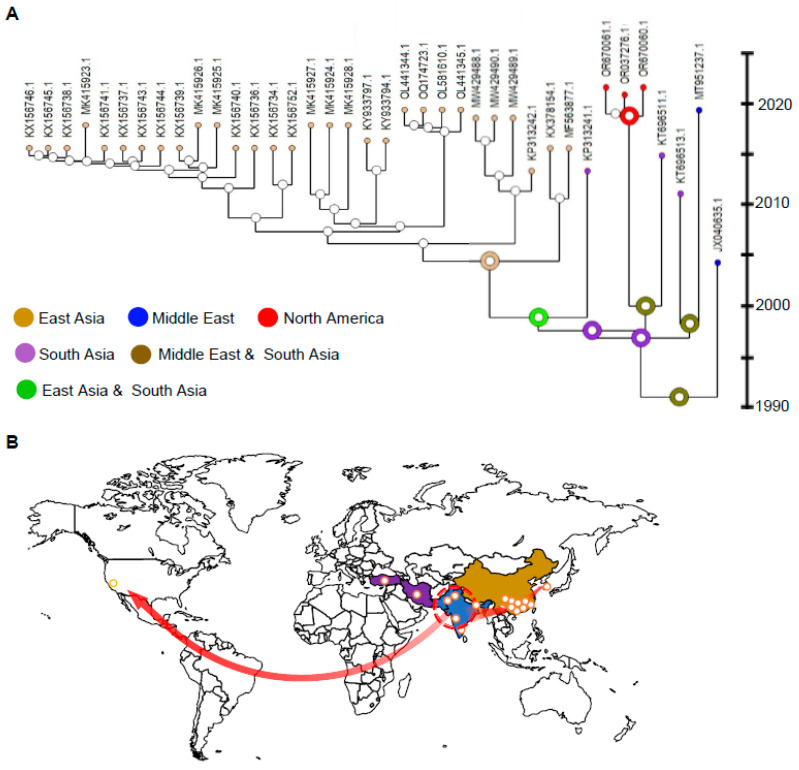
Time-calibrated biogeographic phylogenetic tree of *Citrus yellow vein clearing virus* (CY VCV). (**A**). Cropped reconstruction of CYVCV evolutionary history and dynamics of upon Bayesian evolutionary analysis. Colored dots indicate the biogeographic distribution of CYVCV isolates. The circles illustrate the mostly likely distribution status of the most recent common ancestor in key nodes only. (**B**). A schematic map exhibits the CYVCV outbreaks and hypothesized transmission routes. The sampling localities (colored spots) were categorized into four regions of major clades, as California, USA/North America, South Asia (Pakistan and India shaded in blue), Middle East (Türkiye and Iran shaded in purple), and East Asia (China and South Korea shaded in brown). The red arrows point to the referred transmission routes.

**Table 1 viruses-16-00188-t001:** The group, accession number, name, location, and collected years of 57 reported *Citrus yellow vein clearing virus* (CYVCV) genomes.

Groups	Accession Number	Isolate Name	Location	Collected Year
South Asia, Middle East, and North America	OR037276.1	CYVCV CA1	California, USA	2022
South Asia, Middle East, and North America	OR670060	CYVCV CA2	California, USA	2023
South Asia, Middle East, and North America	OR670061	CYVCV CA3	California, USA	2023
South Asia, Middle East, and North America	KT345342.1	CYVCV IS	Pakistan	1992
South Asia, Middle East, and North America	KP313241.1	CYVCV KP	Pakistan	2012
South Asia, Middle East, and North America	JX040635.1	CYVCV Y1	Türkiye	2000
South Asia, Middle East, and North America	MT951237.1	CYVCV Y2	Türkiye	2020
South Asia, Middle East, and North America	KT696513.1	CYVCV KPMI	India	2009
South Asia, Middle East, and North America	KT696512.1	CYVCV PALI	India	2012
South Asia, Middle East, and North America	KT696511.1	CYVCV RMGI	India	2014
South Asia, Middle East, and North America	KT696510.1	CYVCV ECAI	India	2014
East Asia/C1	NC_026592.1	CYVCV CQ	Chongqing, China	2012
East Asia/C1	MK415926.1	CYVCV HP-MRJ	Yunnan, China	2018
East Asia/C1	MK415925.1	CYVCV JS-RN1	Yunnan, China	2018
East Asia/C1	MK415923.1	CYVCV YN-KPJ	Yunnan, China	2018
East Asia/C1	KX156746.1	CYVCV JX-NF	Jiangxi, China	2015
East Asia/C1	KX156747.1	CYVCV JX-NH	Jiangxi, China	2015
East Asia/C1	KX156745.1	CYVCV HN-STG	Hunan, China	2015
East Asia/C1	KX156744.1	CYVCV HN-GXP	Hunan, China	2015
East Asia/C1	KX156743.1	CYVCV GZ-GXP	Guizhou, China	2015
East Asia/C1	KX156741.1	CYVCV GX-GXP	Guizhou, China	2015
East Asia/C1	KX156740.1	CYVCV GD-STP	Guangdong, China	2015
East Asia/C1	KX156739.1	CYVCV GD-STJ	Guangdong, China	2015
East Asia/C1	KX156738.1	CYVCV GD-JG	Guangdong, China	2015
East Asia/C1	KX156737.1	CYVCV FJ-PK	Fujian, China	2015
East Asia/C1	KX156736.1	CYVCV CQ-TA	Chongqing, China	2015
East Asia/C1	KX156735.1	CYVCV CQ-PO	Chongqing, China	2015
East Asia/C1	KT124646.1	CYVCV HU	Hunan, China	2007
East Asia/C1	KP313240.1	CYVCV CQ	Chongqing, China	2012
East Asia/C2	KP313242.1	CYVCV YN	Yunnan, China	2012
East Asia/C2	KP120977.1	CYVCV RL	Yunnan, China	2009
East Asia/C2	MW429489.1	CYVCV AY142	Sichuan, China	2019
East Asia/C2	MW429487.1	CYVCV AY112	Sichuan, China	2019
East Asia/C2	MG878869.1	CYVCV AY204	Sichuan, China	2012
East Asia/C2	MW429488.1	CYVCV AY132	Sichuan, China	2019
East Asia/C2	MW429491.1	CYVCV AY221	Sichuan, China	2019
East Asia/C2	MW429490.1	CYVCV AY212	Sichuan, China	2019
East Asia/C3	KX156750.1	CYVCV YN-BTC	Yunnan, China	2015
East Asia/C3	KX156752.1	CYVCV YN-NH	Yunnan, China	2015
East Asia/C3	KX156751.1	CYVCV YN-EL	Yunnan, China	2015
East Asia/C3	KX156734.1	CYVCV YN-SA	Yunnan, China	2015
East Asia/C4	KX156742.1	CYVCV GX-STJ	Guangxi, China	2015
East Asia/C4	KX156749.1	CYVCV SC-NH	Sichuan, China	2015
East Asia/C4	KX156748.1	CYVCV SC-EL	Sichuan, China	2015
East Asia/C5	KX378154.1	CYVCV JX	Jiangxi, China	2015
East Asia/C5	MF563877.1	CYVCV JX	Jiangxi, China	2015
East Asia/C6	KY933797.1	CYVCV ZJ4	Zhejiang, China	2016
East Asia/C6	KY933796.1	CYVCV ZJ3	Zhejiang, China	2016
East Asia/C6	KY933795.1	CYVCV ZJ2	Zhejiang, China	2016
East Asia/C6	KY933794.1	CYVCV ZJ1	Zhejiang, China	2016
East Asia/C7	MK415928.1	CYVCV BJ-QPJ	Yunnan, China	2018
East Asia/C7	MK415927.1	CYVCV BJ-WG	Yunnan, China	2018
East Asia/C7	MK415924.1	CYVCV BJ-HMR	Yunnan, China	2018
East Asia/SK1	OL581610.1	CYVCV GJ1	South Korea	2020
East Asia/SK1	OL441344.1	CYVCV GJ2	South Korea	2020
East Asia/SK1	OQ174723.1	CYVCV GJ3	South Korea	2021
East Asia/SK1	OL441345.1	CYVCV GJ4	South Korea	2020

**Table 2 viruses-16-00188-t002:** The group, accession number, location, and collected years of 79 *Citrus yellow vein clearing virus* (CYVCV) coat protein sequences.

Groups	Accession Number	Location	Collected Year
North America	OR670060/CP	USA	2023
North America	OR670061/CP	USA	2023
North America	WKQ63386.1	USA	2022
Middle East	AFO65015.1	Türkiye	2000
Middle East	UIS31351.1	Türkiye	2020
Middle East	MW600321.1	Iran	2021
Middle East	MW600323.1	Iran	2021
Middle East	MW600325.1	Iran	2021
Middle East	MW600324.1	Iran	2021
Middle East	MN547328.1	Iran	2019
Middle East	KX902487.1	Iran	2016
Middle East	MW600322.1	Iran	2021
Middle East	MN547331.1	Iran	2019
Middle East	MN547329.1	Iran	2019
Middle East	MN547330.1	Iran	2019
Middle East	KX902486.1	Iran	2016
Middle East	KX902488.1	Iran	2016
South Asia	ANQ46537.1	Pakistan	1992
South Asia	AJO26409.1	Pakistan	2012
South Asia	AOO32398.1	India	2009
South Asia	KT696514.1	India	2014
South Asia	KT696515.1	India	2014
South Asia	AOO32392.1	India	2012
South Asia	KT696521.1	India	2014
South Asia	KT696517.1	India	2014
South Asia	KT696519.1	India	2014
South Asia	AOO32380.1	India	2014
South Asia	KT696522.1	India	2014
South Asia	KT696516.1	India	2014
South Asia	KT696518.1	India	2014
South Asia	KT696520.1	India	2014
South Asia	AOO32386.1	India	2014
East Asia	QZN83553.1	China	2020
East Asia	QIN54958.1	China	2018
East Asia	QIN54976.1	China	2018
East Asia	AVQ67743.1	China	2016
East Asia	AVQ67749.1	China	2016
East Asia	AVQ67761.1	China	2016
East Asia	AVQ67755.1	China	2016
East Asia	QIN54982.1	China	2018
East Asia	WCR39905.1	China	2019
East Asia	WBG00067.1	China	2019
East Asia	AWJ64286.1	China	2012
East Asia	WBG00073.1	China	2019
East Asia	WBG00079.1	China	2019
East Asia	AKG26810.1	China	2009
East Asia	WBG00085.1	China	2019
East Asia	WBG00091.1	China	2019
East Asia	AJO26415.1	China	2012
East Asia	QIN54952.1	China	2018
East Asia	YP009124992.1	China	2012
East Asia	AJO26403.1	China	2012
East Asia	ANO39204.1	China	2007
East Asia	QIN54970.1	China	2018
East Asia	ASK39436.1	China	2015
East Asia	ASK39538.1	China	2015
East Asia	ASK39520.1	China	2014
East Asia	ASK39526.1	China	2015
East Asia	ASK39532.1	China	2015
East Asia	ASK39484.1	China	2015
East Asia	ASK39544.1	China	2015
East Asia	ASK39442.1	China	2014
East Asia	ASK39448.1	China	2014
East Asia	ASK39472.1	China	2015
East Asia	ASK39466.1	China	2015
East Asia	ASK39460.1	China	2015
East Asia	ASK39502.1	China	2015
East Asia	ASK39508.1	China	2015
East Asia	ASK39478.1	China	2014
East Asia	ASK39514.1	China	2015
East Asia	ASK39454.1	China	2015
East Asia	ASK39496.1	China	2015
East Asia	ASK39490.1	China	2015
East Asia	QIN54964.1	China	2018
East Asia	AWV91523.1	China	2015
East Asia	AOX49234.1	China	2015
East Asia	UYW66709.1	South Korea	2022
East Asia	UYW66724.1	South Korea	2022
East Asia	UYW66718.1	South Korea	2022

## Data Availability

Data are contained within the article and Appendix A. The nucleotide sequences of the complete genome sequence of CYVCV CA1, CA2, and CA3 obtained in this study were submitted to the GenBank database under accession numbers OR37276.1, OR670060.1, and OR670061.1.

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
