# Peer review of "Genotype Sequencing and Phylogenetic Analysis Revealed the Origins of Citrus Yellow Vein Clearing Virus California Isolates"

_viruses, 2024, doi:10.3390/v16020188_

Round 1
Reviewer 1 Report
Comments and Suggestions for Authors
Author Response
We are grateful for the positive feedback provided by the reviewer on our manuscript. Below,please find our responses and the corresponding edits made in accordance with the reviewer's comments.
Introduction section:
The rapid proliferation of CYVCV in China since 2009 has resulted in substantial losses in
lemon production where the disease incidence with high [17].
Please change as follows:
The rapid proliferation of CYVCV in China since 2009 has resulted in substantial losses in
lemon production where the disease incidence was high [17].
The grammar mistake has been modified. “With” was replaced by “Was”.
Fig 1 panel C: What does the Y axis represent? Relative amounts of the virus? Nad5?
The Y axis represents the relative expression of citrus yellow vein clearing virus. In the revised manuscript, we added notes to the Figure 1 legend as “X axis depicts different citrus organs, Y axis stands for the relative expression of Citrus yellow vein clearing virus” to address this comment.
Fig 2. Can be moved to the supplementary files section. It is very difficult to read.
We acknowledge the reviewer's observation regarding the size of Figure 2, and we recognize the difficulty in locating information on citrus yellow vein clearing virus, even when marked in red. We suspect this issue may have arisen during the figure insertion process. To address this concern, we will provide figures in a higher resolution. Moreover, we understand that Figure 2 is directly relevant to the results section, where we highlight the use of long-read sequencing technology to obtain complete genomes of three California CYVCV isolates (CA1, CA2, and CA3). These sequences demonstrated intergenomic similarities ranging from 95.4% to 97.4% to 54 publicly available CYVCV genome sequences, indicating a relatively low level of heterogeneity. Despite its size, we believe it is essential to retain Figure 2 in the main body rather than moving it to the supplementary section.

Reviewer 2 Report
Comments and Suggestions for Authors
CYVCV is characterized by an epidemiological profile that let to envisage it could become the citrus virus more widespread worldwide. The paper contributes to increase the knowledge on the spread in California and supplies more information on phylogenetical and taxonomic aspects.
It would be improved by minor revisions.
- Introduction: should be enforced with information about the alternative hosts reported in some countries. It is important for the epidemiology of such a virus ?
- Genome sequencing: there is an advantage on the use of long reads technology preferred for a quite short virus ?
- There are other information that could be added to the complete lists of genome and CP sequences presented in the tables to gain other elements of comparison? (i.e. the plant species hosting the virus sequenced?... the number and size of the reads).
- The interesting graphs in the figures look not clear wonder if it is a technology inconvenient or the complex amount of details. A simplification would be useful. Sometime is less attractive to have a large figure to show a single point of connection.
Author Response
Responses:
We express our gratitude to the reviewer for the positive feedback. In response to their comments, we have made revisions to the manuscript. Please find the modifications outlined below.
- Introduction: should be enforced with information about the alternative hosts reported in some countries. It is important for the epidemiology of such a virus ?
We concur with the reviewer's suggestion to incorporate information about alternative hosts into the introduction. Accordingly, we have appended a few sentences to the end of the first paragraph, stating, 'In addition to infecting most citrus species, cultivars, and hybrids, CYVCV has been reported to transmit to some non-citrus hosts, including wild grapevine (Vitis vinifera), common bean (Phaseolus vulgaris), cowpea (Vigna unguiculata), common mallow (Malva sylvestris), quinoa (Chenopodium quinoa), and some other herbaceous species [4-6].
- Genome sequencing: there is an advantage on the use of long reads technology preferred for a quite short virus ?
Long-read sequencing surpasses traditional short-read methods by enabling the sequencing of significantly longer DNA fragments. This capability empowers the detection of intricate structural variants that might pose challenges for short-read methods. Furthermore, in the case of the relatively short 7.5 KB citrus yellow vein clearing virus, long-read sequencing eliminates the necessity for assembly programs, thereby saving valuable time.
There are other information that could be added to the complete lists of genome and CP sequences presented in the tables to gain other elements of comparison? (i.e. the plant species hosting the virus sequenced?... the number and size of the reads).
We concur with the reviewer's suggestion to enhance the table by including additional details such as the plant species hosting the sequenced virus and the corresponding number and size of reads. Our comparison revealed that, regrettably, certain isolates lack this information in the data downloaded from the public library.
- The interesting graphs in the figures look not clear wonder if it is a technology inconvenient or the complex amount of details. A simplification would be useful. Sometime is less attractive to have a large figure to show a single point of connection.
We appreciate the reviewer's feedback regarding the clarity of the figures in the PDF version. We acknowledge that the figures may not be clear, and we suspect this issue may have occurred during the figure insertion process. To ensure optimal visual quality, we will submit figures to the Viruses editors in a higher resolution for the final publication.
